# Redox-Active Monolayers Self-Assembled on Gold Electrodes—Effect of Their Structures on Electrochemical Parameters and DNA Sensing Ability

**DOI:** 10.3390/molecules25030607

**Published:** 2020-01-30

**Authors:** Kamila Malecka, Shalini Menon, Gopal Palla, Krishnapillai Girish Kumar, Mathias Daniels, Wim Dehaen, Hanna Radecka, Jerzy Radecki

**Affiliations:** 1Institute of Animal Reproduction and Food Research, Polish Academy of Sciences, Tuwima 10, 10-748 Olsztyn, Poland; k.malecka@pan.olsztyn.pl (K.M.); palla.gopal@pan.olsztyn.pl (G.P.); h.radecka@pan.olsztyn.pl (H.R.); 2Department of Applied Chemistry, Cochin University of Science and Technology, Kochi, Kerala 682022, India; salini.smn@gmail.com (S.M.); drkgirish@gmail.com (K.G.K.); 3Molecular Design and Synthesis, Department of Chemistry, KU Leuven, Leuven Chem&Tech, Celestijnenlaan 200F, B-3001 Leuven, Belgium; mathias.daniels@kuleuven.be (M.D.); wim.dehaen@kuleuven.be (W.D.)

**Keywords:** synthetic ligands—transition metal complexes, electrochemical properties, DNA sensing

## Abstract

The background: The monolayers self-assembled on the gold electrode incorporated transition metal complexes can act both as receptor (“host” molecules) immobilization sites, as well as transducer for interface recognitions of “guest” molecules present in the aqueous solutions. Their electrochemical parameters influencing the sensing properties strongly depend on the transition metal complex structures. The objectives: The electrochemical characterization of the symmetric terpyridine–M^2+^–terpyridine and asymmetric dipyrromethene–M^2+^–terpyridine complexes modified with ssDNA probe covalently attached to the gold electrodes and exploring their ssDNA sensing ability were the main aims of the research presented. The methods: Two transition metal cations have been selected: Cu^2+^ and Co^2+^ for creation of redox-active monolayers. The electron transfer coefficients indicating the reversibility and electron transfer rate constant measuring kinetic of redox reactions have been determined for all SAMs studied using: Cyclic Voltammetry, Osteryoung Square-Wave Voltammetry, and Differential Pulse Voltammetry. All redox-active platforms have been applied for immobilization of ssDNA probe. Next, their sensing properties towards complementary DNA target have been explored electrochemically. The results: All SAMs studied were stable displaying quasi-reversible redox activity. The linear relationships between cathodic and anodic current vs. san rate were obtained for both symmetric and asymmetric SAMs incorporating Co^2+^ and Cu^2+^, indicating that oxidized and reduced redox sites are adsorbed on the electrode surface. The ssDNA sensing ability were observed in the fM concentration range. The low responses towards non-complementary ssDNA sequences provided evidences for sensors good selectivity. The conclusions: All redox-active SAMs modified with a ssDNA probe were suitable for sensing of ssDNA target, with very good sensitivity in fM range and very good selectivity. The detection limits obtained for SAMs incorporating Cu^2+^, both symmetric and asymmetric, were better in comparison to SAMs incorporating Co^2+^. Thus, selection of the right transition metal cation has stronger influence on ssDNA sensing ability, than complex structures.

## 1. Introduction

The discovery of self-organization of molecules, for which has been awarded the Nobel Prize in 1987 to Lehn, Pedersen and Cram has opened the gate for the development of supramolecular chemistry. It is a truly interdisciplinary science concerning chemical, physical and biological properties of molecules making them organized by means of intermolecular (non-covalent) binding interactions. The numerous perspectives of supramolecular chemistry have been wonderfully presented by Jean -Marie Lehn [1]. A further great discovery by Allara and Nuzzo on self-organization of molecules possessing thio-functional groups on the surface of noble metals (Au, Pt, Ag) has been seminal to the exploration of supramolecular interactions at the interphase [2,3]. This approach allows for the creation of complex structures using synthetic compounds in order to explore biological processes occurring at the interface between aqueous media and the solid surface of an electrode, with excellent selectivity and sensitivity.

Being involved in molecular recognition research, we have developed electrochemical sensing platforms incorporating redox-active centers complexed by organic ligands [4,5] The redox-active sites play a double role. They act as the binding units for the specific host molecule responsible for the analyte (guest) recognition at the electrode surface. Also, they are responsible for processing the energy generated by the host-guest interactions into a readable analytical signal. Such sensing systems possess substantial advantages. The modification layers could be created directly on the electrode surface. This allows to eliminate synthetic work and reduce the consumption of chemicals. The host—guest interactions occurring at the interphase are directly converted into an electronic signal. There is no need to apply redox-active markers in the sample solution. Such systems need a very low, in µL level, sample volume. In general, they are selective and very sensitive.

Redox-active layers involving dipyrromethene–M^2+^–dipyrromethene or (phenantroline)_3_–Fe^2+^ complexes have been successfully applied for the sensitive and selective determination of DNA and RNA targets [6,7,8]. The main goal of the research presented is the elaboration of the relationship between the chemical structure of the redox-active layer and the amount of generated analytical signal. Therefore, a series of electrochemically active layers: terpyridine–M^2+^–terpyridine (TPY-M^2+^-TPY) and dipyrromethene–M^2+^–terpyridine (DPM-M^2+^-TPY) (M^2+^: Co^2+^ and Cu^2+^) deposited on the surface of gold via Au-S covalent bonds have been explored (Scheme 1). These redox-active complexes have been applied for covalent immobilization of ssDNA probe responsible for recognition of complementary target ssDNA derived from Avian influenza virus H5N1.

The key impact of the research is the electrochemical exploring of redox-active SAMs having different structures and different metal centers and their sensing ability of complementary ssDNA.

## 2. Results

### 2.1. Electrochemical Characterization of Gold Electrodes Modified with Symmetric TPY/M(II)/TPY Complexes

Each step of the gold electrode modification, illustrated in Scheme 1A, has been controlled using cyclic voltammetry (CV) and Osteryoung square wave voltammetry (OSWV). Terpyridine incorporated into MBL-TPY-mixed SAM showed electrochemical activity. The reduction and oxidation peaks were observed at 272 and 317 mV, respectively. The complexation of Co(II) caused an increase of reduction and oxidation current with very small peak potential shifts in the lower value direction (Figure 1A, Table 1). The complexation of Cu(II) had a stronger influence on the redox properties of TPY SAM. A substantial increase of the reduction and oxidation current were observed. Both peak potentials gained lower values in comparison to TPY SAM (Figure 1B, Table 1). The influence of Co(II) and Cu(II) complexation on the redox properties of TPY SAM has been explored also by OSWV. The current increase, assisted with a peak potential shift, was larger for TPY/Cu(II) in comparison to TPY/Co(II) (Figure 2A,B; Table 1). 

In order to get saturation of the coordination sphere of the Co(II) and Cu(II) centers, TPY-NHS was used. Next, via an NHS group NH_2_-(CH_2_)_2_-OH (EA) or NH_2_-ssDNA were attached to the TPY-M(II)-TPY complexes. The TPY-Co(II)-TPY SAM functionalized with EA or ssDNA showed a lower reduction current assisted with peak potential shifts towards a higher value in the comparison with a TPY-Co(II) SAM. The oxidation current values were very similar, but the potential of the oxidation peaks was shifted to higher values (Appendix A; Table 1). When Cu(II) centers were incorporated into the SAMs, the reduction current of SAMS functionalized with EA as well as ssDNA were very similar to the value obtained for TPY-Cu(II) SAM, but the potential of the reduction peaks was shifted to a higher value. On the other hand, the potential of the oxidation peaks for these SAMs remained unchanged, while the oxidation current was substantially lower in comparison to the value obtained for DPM-Cu(II) SAM (Appendix A, Table 1). Using OSWV technique, which eliminates capacitive current, a lower current was observed for TPY-M(II)-TPY-ssDNA SAMs than for TPY-M(II)-TPY-EA SAMs. This difference was larger when Co(II) was applied (Appendix A; Table 1).

Cyclic voltammetry at different scan rates was performed for the electrodes modified with TPY-M(II)-TPY-EA and TPY-M(II)-TPY-ssDNA. The linear relationship of reduction and oxidation current vs. scan rate confirms the presence of Co(II) and Cu(II) centers on the surface of the gold electrodes (Appendix A). The slope of the cathodic peak current obtained for TPY-Co(II)-TPY-ssDNA was larger than the slope recorded for TPY-Co(II)-TPY-EA (Appendix A). The same tendency was observed for Cu(II) SAMs. However, the values of the slopes were substantially smaller than those obtained for Co(II) SAMs (Appendix A).

### 2.2. Electrochemical Characterization of Gold Electrodes Modified with Asymmetric DPM/M(II)/TPY Complexes

The electrochemical parameters obtained for asymmetric redox-active SAMs consisting of DPM/M(II)/TPY complexes (Scheme 1B) were collected in Table 2. The dipyrromethene incorporated into MBL-DPM SAM showed quasi-reversible electrochemical activity with reduction and oxidation peaks at 252 and 316 mV, respectively. The complexation of Co(II) did not influence substantially on the DPM redox behavior (Figure 1C, Table 2). The incorporation of Cu(II) into the DPM SAM results in a substantial increase of the reduction and oxidation current assisted with a reduction peak potential shift to lower values, and an oxidation peak potential shift to higher values (Figure 1D, Table 2). A slight increase of the current and peak potential shift was observed for DPM-Co(II) using OSWV (Figure 2C, Table 2). The complexation of Cu(II) caused a substantial redox current increase, with almost no potential change (Figure 2D, Table 2). The DPM-M(II) SAMS, after complexation with TPY-NHS, were functionalized with EA or ssDNA. The saturation of coordination sphere of DPM-Co(II) and DPM-Cu(II) centers with TPY-NHS caused a reduction current decrease assisted by potential shifts to higher values. The changes of these two parameters were larger for DPM-M(II)-TPY-ssDNA than for DPM-M(II)-TPY-EA (Table 2, Appendix A). The oxidation current recorded for DPM-M(II)TPY-EA and DPM-M(II)TPY-ssDNA was lower than the values obtained for DPM-M(II) SAMs. The oxidation peak potentials were shifted to the lower values (Table 2). After elimination of capacitive current using OSWV, a lower redox current was recorded for DPM-M(II)-TPY-ssDNA in comparison to the current obtained for DPM–M(II)–TPY–EA and this was clearly visible for Co(II) and Cu(II) redox-active SAMs (Table 2, Appendix A).

The asymmetric DPM-M(II)-TPY SAMs were stable, allowing one to perform cyclic voltammetry at different scan rates. The linear relationships between oxidation and reduction current vs. scan rate confirm the immobilization of Co(II) as well as Cu(II) redox centers on the electrode surface (Appendix A). The slopes of the reduction current vs. the scan rate were higher than the slopes of the oxidation current vs. scan rate for Co(II) SAMs (Appendix A). The opposite tendency was observed for Cu(II) SAMs. The slopes of the oxidation current vs. the scan rates were larger than the slopes of the reduction current vs. the scan rate (Appendix A). The values of the slopes for reduction and oxidation current vs. the scan rate obtained for DPM-M(II)-TPY–ssDNA were higher than those recorded for DPM-M(II)-TPY–EA (Appendix A). Thus, it might be concluded that the functional groups influence the redox properties of asymmetric SAMs.

### 2.3. Comparison of Electrochemical Parameters of Symmetric TPY/M(II)/TPY and Asymmetric DPM/M(II)/TPY Sensing Platforms Functionalized with Ethanolamine and ssDNA

The values of the electron transfer coefficient *α* and the electron transfer rate constant *k* [s^−1^] were calculated based on the generally applicable Laviron’s procedure (Appendix A) [9]. The values of electron transfer coefficient (α) as well electron transfer rate constants (*k*) obtained for redox-active SAMs incorporating Co(II) centers were > 0.5. This indicated that for these SAMs the reduction process is favorable. The *α* values < 0.5 obtained for SAMs with Cu(II) centers pointed out that for this transition metal cation the oxidation process is favorable (Table 3) [10]. The replacement of EA with ssDNA caused an increase of α as well as *k* values, indicating improvement of electronic contact between the electroactive centers and electrode surface, which facilitates cations accessibility from the bulk solution to redox centers in order to balance the charge generated due to the redox processes [6]. The probable reason for this phenomenon is the negative charges of the ssDNA strand.

From the scan rate studies performed for both symmetric and asymmetric SAMs incorporating Co(II) and Cu(II), linear relationships between cathodic and anodic current vs. scan rate were obtained. This indicated that oxidized and reduced redox sites are adsorbed on the electrode surface. Based on the slopes of the relationships presented in Appendix A, the surface coverage (*Γ*) values were calculated using the following Equation (1) [11]: (1)$$i_{p}=\;\frac{n^{2}F^{2}\nu A\Gamma }{4RT},$$
where ‘*n*’ is the number of electrons involved in the oxidation or reduction process, ‘*F*’ is the Faraday’s constant, ‘ν’ is scan rate, ‘*A*’ is the area of the electrode, ‘*R*’ is ideal gas constant, ‘*T*’ is the temperature and ‘*Γ*’ is the surface coverage of the adsorbate.

For SAMs incorporating Co(II) the *Γ* values obtained based on the reduction current were larger than the values obtained based on the oxidation current (Table 4; Appendix A). Thus, it might be concluded that Co(II) is a more favorable oxidation state than Co(III). The opposite phenomenon was observed for SAMs incorporating Cu(II). For these SAMs larger *Γ* values were obtained based on oxidation current (Table 4, Appendix A). These data confirm that Cu(II) is a more favorable oxidation state than Cu(I) ensuring the stronger adsorption. The comparison of *Γ* values obtained for symmetric and asymmetric SAMs clearly indicated that TPY-M(II)-TPY redox-active complexes have higher adsorption capability than DPM-M(II)-TPY complexes (Table 4).

### 2.4. Electrochemical Determination of Target ssDNA Using Gold Electrodes Modified with Symmetric TPY/M(II)/TPY-ssDNA and Asymmetric DPM/M(II)/TPY-ssDNA Complexes

The redox-active SAMs, symmetric and asymmetric, incorporating Co(II) or Cu(II) sites (Scheme 1), were used for sensing target ssDNA derived from Avian influenza viruses H5N1. A decrease of the redox current measured with OSWV in the presence of complementary ssDNA (c-NC3) to the ssDNA probe was observed for all sensing platforms (Figure 3 and Figure 4). More visible potential shifts in the presence of complementary DNA were observed for sensing SAMs (symmetric and asymmetric) incorporating Co(II). 

All systems studied were selective. The non-complementary ssDNA (nc-NC3) generated only negligible responses (Figure 3 and Figure 4). The selectivity was calculated based on the slope ratio as recommended by Wang and Umezawa [12,13], Equation (2):
*R*_i,j_ = *S*_j_/*S*_i_,(2)
where: *S_i_* is the slope of the calibration curve for complementary ssDNA; *S_j_* is the slope of the calibration curve for non-complementary ssDNA. 

The low values of *R_i,j_* confirmed the very good selectivity of all sensing platforms studied. The most selective was the electrode modified with TPY-Cu(II)-TPY-ssDNA (Table 5).

All sensing platforms were very sensitive. The linear relationships between values of current decrease and concentration of target ssDNA was observed from 1.0 to 20.0 fM (Figure 5). The detection limit (LOD) for each sensor was calculated using Equation (3), where *SD* is the standard deviation of the lowest response and *m* is the slope of the calibration curve [14]:
LOD = (3.3 × SD)/m(3)

LODs for all sensing platforms were at the fM level. However, LODs obtained for platforms incorporating Cu(II) centers were superior in comparison to the values obtained for the platforms incorporating Co(II) centers (Table 5). The redox current observed for symmetric and asymmetric Cu(II) SAMs functionalized with ssDNA was higher than in the case of Co(II) SAMs (Table 1 and Table 2; Appendix A). This could be the reason of the better detection limits of the Cu(II) platforms, in comparison to the Co(II) platforms. 

## 3. Discussion

Terpyridine, having three strong nitrogen coordination sites is an excellent ligand for chelation with various metal cations. The three electron-deficient pyridines make terpyridine a strong σ-donor and π-receptor. In addition, the low LUMO of terpyridine, allows this ligand to participate in redox reactions as a “non-innocent“ molecule [15]. The “close shell” TPY-M(II)-TPY complexes have been widely applied by the Nishihara group for the creation of supramolecular structures of π-conjugated redox molecular wires [16,17]. The redox activity of (TPY)_2_–Co(II) deposited on the electrode surface has been proved by Abruna and Forster [18,19]. Therefore, we have selected this ligand for fabrication of redox-active platforms for sensing ssDNA. The redox-active layers involving dipyrromethene-M^2+^-dipyrromethene have been already successfully applied by us for the sensitive and selective determination of ssDNA target [6]. In this sensing layer, the redox centers, responsible for analytical signal transduction, are located very close to the electrode surface. Therefore, their positions remain unchanged after creation of dsDNA. Nevertheless, a decrease of the redox Cu(II) and Co(II) current was observed after the hybridization processes occurring on the electrode surface. The reduction and oxidation processes of metal centers cause a change of their charge. Their neutralization by the ions present in the supporting electrolyte is a condition allowing to proceed with the redox reactions. The creation of dsDNA changed the accessibility of ions to the redox centers. This new type of analytical signal generation was named “the ion-barrier switch off” [6].

The new redox-active layers presented here, destined for ssDNA sensing, are based on the same mechanism. Two kinds of complexes have been studied, symmetric TPY-M(II)-TPY, and asymmetric DPM-M(II)-TPY. The redox activities of TPY and DPM SAMs have been substantially changed after cation complexation, in particular, Cu(II) (Figure 1). The type of metal center showed a stronger influence on the redox properties of SAMs studied in the comparison to the influence of the ligand structure (Figure 1 and Figure 2, Table 1 and Table 2). It has been proved that all type of redox-active SAMs are stable. The metal centers, at both oxidation states, were adsorbed on the electrode surface (Table 4, Appendix A). Thus, it might be concluded that all platforms are suitable for sensing of ssDNA. This was confirmed and the results are presented in Figure 3, Figure 4 and Figure 5. All modified electrodes were very selective and displayed very good sensitivity in the fM range. Only slightly better detection limits were obtained for SAMs, symmetric and asymmetric, incorporating Cu(II) units (Table 5). 

Thus, the importance of the right selection of the cation is the main conclusion of the research presented concerning the influence of the structure of redox-active complexes, deposited on a gold electrode surface, on their DNA sensing ability.

Our sensing approaches are superior to most reported sensors in terms of LOD (Appendix A) with only two exceptions. The lower LOD values were achieved due to incorporation of NH_2_-3-iron bis(dicarbollide)-ssDNA probes [SI3] or the silver nanoclusters [SI 9]. However, it could underline that sensors presented possess simple fabrication procedure as well as they have very universal character. The redox-active platforms proposed could be applied for immobilization of any ssDNA probes.

## 4. Materials and Methods

**Reagents and Materials.** The thiol derivative of terpyridine–NHS (TPY–NHS) was synthesized at the Chemical Department of Leuven University (Appendix A). The thiol derivative of dipyrromethene (DPM-SH) was synthesized by ProChimia Surface Company (Sopot, Poland). 6-Amino-1-hexanethiol hydrochloride (AHT), 4-Mercapto-1-butanol (MBL), copper (II) acetate Cu(OAc)_2_, cobalt (II) acetate Co(OAc)_2_, 2-morpholinoethanesulfonic acid (MES), ethanolamine (EA) and dichloromethane (DCM) and phosphate buffer saline (PBS, pH 7.4) components (137 mM sodium chloride NaCl, 2.7 mM potassium chloride KCl, 10 mM Potassium phosphate monobasic KH_2_PO_4_, 1.8 mM Sodium phosphate dibasic Na_2_HPO_4_) were purchased from Sigma–Aldrich (Poznań, Poland). Potassium hydroxide, sulfuric acid, hydrogen peroxide, ethanol (EtOH) and methanol (MeOH) were obtained from Avantor Performance Materials (Gliwice, Poland). The modified oligonucleotide NH_2_-ssDNA (5′-NH_2_-(CH_2_)_6_-CCT CAA GGA GAG AGA AGA AG-3′) was applied as a probe (named NH_2_-NC3) for immobilization on the surface of gold electrodes, while two unmodified oligonucleotides, c-NC3 (5′-CTT CTT CTC TCT CCT TGA GG-3′) and nc-NC3 (5′-GAA GAA GAG AGA GGA ACT CC-3′) served as complementary and non-complementary hybridization targets, respectively. These oligonucleotides were supplied by Biomers (Ulm, Germany). 

All aqueous solutions were prepared with deionized and charcoal-treated water (resistivity of 18.2 MΩ·cm) purified with a Milli-Q reagent grade water system (Millipore, Bedford, MA, USA). All solutions were deoxygenated by purging with nitrogen (ultra-pure 6.0, Air Products, Warsaw, Poland) for 15 m. PBS was autoclaved before use. 

**Electrochemical Measurements.** All electrochemical measurements were performed with a potentiostat-galvanostat AutoLab (Eco Chemie, Utrecht, The Netherlands) with a three-electrode configuration system. Potentials were measured *versus* the silver chloride (Ag/AgCl) electrode, and a platinum wire was used as an auxiliary electrode. The voltammetric experiments were carried out in an electrochemical cell of 5 mL volume.

Cyclic voltammetry (CV) measurements were performed in the potential range from 750 mV to –200 mV for the gold electrodes modified with the Cu(II) complexes and from –200 mV to 750 mV for the gold electrodes modified with the Co(II) complexes with different scan rate 10, 50, 100, 200, 300, 400, 500, 600, 700, 800, 900 and 1000 mVs^−1^. Osteryoung square wave voltammetry (OSWV) was performed with a potential from 700 mV to –100 mV for the gold electrodes modified with the Cu(II) complex and from –100 mV to 700 mV for the gold electrodes modified with the Co(II) complex with a step potential of 1 mV, a square-wave frequency of 50 Hz, and an amplitude of 50 mV. Differential Pulse Voltammetry (DPV) measurements were performed in two cycles: with the potential from –200 mV to 750 mV for oxidation of Co(II) complexes and from 750 mV to –200 mV for reduction of Co(II) complexes; with the potential from 750 mV to –200 mV for reduction of Cu(II) complexes and from –200 mV to 750 mV for oxidation of Cu(II) complexes. The values of the modulation amplitude and step potential were 25 mV and 10 mV, respectively. 

All experiments were carried out at room temperature, in the presence of PBS purged with nitrogen for 15 min. A gentle nitrogen flow was applied over the sample solution during all measurements.

The electrochemical characterization of the electroactive monolayers TPY/Me(II)/TPY/EA, DPM/Me(II)/TPY/ssDNA, TPY/Me(II)/TPY/EA and DPM/Me(II)/TPY/ssDNA, Me(II)=Co(II) or Cu(II), was carried out in the presence of PBS supporting electrolyte using CV, OSWV and DPV techniques.

**Hybridization of the NH_2_-NC3 probe and the target sequences.** The target oligonucleotides (c-NC3, and nc-NC3) were diluted in the PBS pH 7.4 hybridization buffer to the concentration of 1, 5, 10, 15 and 20 fM. Hybridization reactions were performed by dropping 5 μL of the solution containing targets: 20-mer c-NC3 or nc-NC3 in buffer solution as described above, for 30 min at room temperature on the TPY/Me(II)/TPY/ssDNA or DPM/Me(II)/TPY/ssDNA modified gold electrodes. After the hybridization process with the particular concentration of targets, the electrodes were rinsed thoroughly with PBS, immersed in PBS and gently vortexed for 20 s and then rinsed thoroughly again. Next, the electrodes were transferred to the electrochemical cell for the electrochemical measurements. In addition, before starting each experiment the stability of the DNA biosensor was checked by dropping 5 μL of pure PBS buffer solution and equilibrating for 30 min on the modified gold electrodes surfaces.

The hybridization processes were monitored using OSWV with a potential from 750 mV to –100 mV for the gold electrodes modified with the Cu(II) complex and from –100 mV to 750 mV for the gold electrodes modified with the Co(II) complex with a step potential of 1 mV, a square-wave frequency of 50 Hz, and an amplitude of 50 mV in PBS pH 7.4. The electrode responses were expressed as: I = [(I_n_ − I_0_)/I_0_] × 100%, where I_n_ is the peak current measured in the presence of the analyte and I_0_ the peak current before applying the analyte (in pure buffer).

**Successive steps of genosensor fabrication.** Gold disk electrodes with a radius of 1 mm (Bioanalytical Systems (BASi), West Lafayette, IN, USA) were used for all experiments. The electrodes preparation was performed according to an already published protocol [20]. Briefly, gold disk electrodes (Bioanalytical Systems (BAS), West Lafayette, IN, USA) with a radius of 1 mm were used in all experiments. These electrodes were initially cleaned mechanically by polishing with alumina slurries (Alpha and Gamma Micropolish, Buehler; Lake Bluff, IL, USA). The polished electrodes were subsequently cleaned electrochemically by cyclic voltammetry in KOH (0.5 M) and then in H_2_SO_4_ (0.5 M). Before modification, the electrode surfaces were activated in KOH (0.5 M). After electrochemical cleaning, each electrode was washed with Milli-Q water and stored in water (for several minutes, until the next step) to avoid contamination from air.

Directly after cleaning, the electrodes were rinsed repeatedly with Milli-Q water, MeOH and a mixture of MeOH and DCM (1:1, volume ratio) and dipped in the modification solutions. Preparation of the genosensors has been illustrated in Scheme 1 by the following steps:

(A)

The spontaneous self-assembly of 0.01 mM AHT and 1 mM MBL layer formation—3 h, room temperature (RT), DCM: MeOH (1:1, *v*/*v*); the electrodes were immersed in tubes containing 180 µL of modifying solution; after modification electrodes were carefully rinsed with a mixture of DCM: MeOHReaction between the amine groups of AHT and NHS of 0.1 mM TPY-NHS—1h, RT, DCM: MeOH (1:1); the electrodes were immersed in tubes containing 180 µL of modifying solution; after modification electrodes were carefully rinsed with a mixture of DCM: MeOHComplexation of 1mM Me (II) (Co (II) or Cu (II)) metal ions by 0.1 mM TPY-NHS—1h, RT, DCM: MeOH (1:1); the electrodes were immersed in tubes containing 180 µL of modifying solution; after modification electrodes were carefully rinsed with a mixture of DCM: MeOHThe closure of the coordination sphere of the Me (II) metal ions by 0.1 mM TPY-NHS—1h, RT, DCM: MeOH (1:1 volume ratio); the electrodes were immersed in tubes containing 180 µL of modifying solution; after modification electrodes were carefully rinsed with: a mixture of DCM: MeOH, MeOH, sterilized Milli-Q water and MES buffer pH 7.0Reaction between amine groups of 10 μM NH_2_-NC3 probe and NHS of 0.1 mM TPY-NHS—1h, RT, MES pH 7.0, the electrodes were fixed upside down, 5 µL droplets of solution were spotted on each surface and the electrodes were covered with tubes; after modification electrodes were carefully rinsed with MES pH 7.0 and PBS pH 7.4Deactivation of unbound NHS groups with 1 M solution of EA, 10 min PBS pH 7.4, the electrodes were fixed upside down, 5 µL droplets of solution were spotted on each surface and electrodes were covered with tubes.

(B)

The spontaneous self-assembly of 0.01 mM DPM-SH and 1 mM MBL layer formation—3 h, RT; the electrodes were immersed in tubes containing 180 µL of modifying solution; after modification electrodes were carefully rinsed with a mixture of DCM: MeOHComplexation of 1 mM M (II) (Co (II) or Cu (II)) metal ions by 0.1 mM TPY-NHS—1 h, RT; the electrodes were immersed in tubes containing 180 µL of modifying solution; after modification electrodes were carefully rinsed with a mixture of DCM: MeOHThe closure of the coordination sphere of the M(II) metal ions by 0.1 mM TPY-NHS—1 h, RT; DCM: MeOH (1:1) the electrodes were immersed in tubes containing 180 µL of modifying solution; after modification electrodes were carefully rinsed with: a mixture of DCM: MeOH; MeOH, sterilized Milli-Q water and MES buffer pH 7.0Reaction between amine groups of 10 μM NH_2_-NC3 probe and NHS of 0.1 mM TPY-NHS—1 h, RT; MES pH 7.0; the electrodes were fixed upside down, 5 µL droplets of solution were spotted on each surface and the electrodes were covered with tubes; after modification electrodes were carefully rinsed with MES pH 7.0 and PBS pH 7.4Deactivation of unbound NHS groups with 1 M solution of EA, 10 min, PBS pH 7.4, RT; the electrodes were fixed upside down, 5 µL droplets of solution were spotted on each surface and the electrodes were covered with tubes.

Finally, electrodes were rinsed with PBS pH 7.4, then dipped in this buffer and stored for 38 h at RT.

The concentrations of modification solutions and time of modifications were selected experimentally.

## 5. Conclusions

Two kinds of redox-active layers, symmetric (TPY-M(II)-TPY) and asymmetric (DPM-M(II)-TPY), displayed different redox parameters. More substantial differences were observed for SAMs incorporating Co(II) or Cu(II). The complex structures play a secondary role. All redox-active SAMs modified with an ssDNA probe were suitable for sensing of ssDNA target, with very good sensitivity in fM range and very good selectivity. The detection limits obtained for SAMs incorporating Cu(II), both symmetric and asymmetric, were better in comparison to SAMs incorporating Co(II).

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
