# Peer review of "Redox-Active Monolayers Self-Assembled on Gold Electrodes—Effect of Their Structures on Electrochemical Parameters and DNA Sensing Ability"

_molecules, 2020, doi:10.3390/molecules25030607_

Round 1
Reviewer 1 Report
In this manuscript, terpyridine - M2+ - terpyridine and dipyrromethene – M2+ - terpyridine (M2+: Co2+ and Cu2+) is attached to the gold electrode, and the DNA sensing response of the electrode is studied. Since the manuscript lacks scientific explanations, I would not recommend publishing this paper in the current form.
Comment 1: Page 2 (line 61-62) “Therefore, a series of electrochemically active layers deposited on the surface of gold via Au-S covalent bonds have been explored (Scheme I). “ Since electrochemical deposition highly depends on the applied potential, time, and concentration of electrolyte. I would recommend adding an experimental section with a detail description of the fabrication steps.
Comment 2: I would suggest a detail characterization of the sensing material after each fabrication step.
Comment 3: The standard reduction potential of the copper (II) is higher than that of Cobalt (II). Table I in the manuscript shows the opposite change in the potential. I would suggest explaining the possible reasons behind it.
Comment 4: I would recommend comparing the sensitivity and selectivity using a similar kind of system and include them in table 5 of the manuscript.
Comment 5: Comment 5: I would suggest studying in-depth materials characterization of the electrode. The formation of the composite material should be demonstrated using proper characterization tools. Also, surface morphology plays a significant role in sensing repose.
Reviewer 2 Report
The manuscript describes the development of electrochemical genosensors based on redox-active layers incorporating terpyridine - M2+ - terpyridine and dipyrromethene – M2+ - terpyridine (M2+ : Co2+ and Cu2+) formed step by step on Au electrode surfaces applied for covalet immobilization of a ssDNA probe responsible for recognition of complementary target ssDNA derived from Avian influenza virus H5N1. A linear dynamic range was observed from 1 to 20 fM. In addition, the detection limits obtained for SAMs incorporating Cu(II), both symmetric and asymmetric, were better in comparison to SAMs incorporating Co(II). Although the novelty of the work is limited, the developed genosensors demonstrate an attractive performance and is competitive in terms of sensitivity in fM range and selectivity. Moreover, the paper is well structured. Therefore, this paper may be published in this Journal after addressing the following issues:
Abstract
Q1). A short description of the interpretation/conclusion is missing in the abstract. A good abstract should be one paragraph which summarizes the background, objectives, methods,results and conclusion of the paper.
Introduction
Q2). In the manuscript a discussion on the advantage of the current assay compared with other approaches is missing.
As well as the novelty of the work should be highlighted (What is the key impact of the research?)
Q3). Readers often study tables and figures before they read the text. In the present manuscript, the large tables make it difficult the analysis of results. Just make sure that readers can easily follow the flow of information!
Discussion
Q4). A comparison of the performance of the pruposed redox active layers with other reported systems destined for ssDNA sensing based or not on the mechanism should be added. In addition, presenting the information in a tabel format makes the data much easier to follow.
Author Response
Please, see the attachment

Reviewer 3 Report
The manuscript of Malecka et al describes experimental work on the utilization of surface tethered metal complexes for the detection of DNA hybridization. Single-stranded probe DNA is covalently attached to previously formed monolayers of metal complexes. The pairing of probe and analyte DNA influences the charge transfer properties of the Cu2+ and Co2+ ions, this is why they can act as a transducer. This report is a continuation of a similar related work, where variations of different metal complexes were used. Nevertheless, the present manuscript presents some advancements since the authors could achieve detection limits in the range of fM.
Overall, the manuscript reports on a carefully performed electrochemical study in which an already existing detection scheme is varied and the detection limits are pushed to very low limits. The conclusions are in parts very vague or missing. The control experiments are in general appropriate. Therefore I recommend publication of the manuscript after major revision.
Major criticism:
The main deficit of this manuscript is the lack of a critical comparison of the advantages and disadvantages of the here reported metal complexes in comparison to the previously reported systems by the same group as well as others. The reader expects some orientation on the stability of the metal complexes, the challenges in preparations, and the detection characteristics. In the following, the outcome of this discussion should become part of the abstract, introduction, discussion, and conclusion.
The authors define a “dynamic linear range” however I am wondering about which criteria this range is defined. It seems as the “dynamic linear range” is identical to the tested concentration range. However, a much larger concentration range than the linear range needs to be investigated to be able to identify the true “dynamic linear range” of a proof of concept sensor, where the saturation of the sensor signal can be observed. Furthermore, I would recommend using the IUPAC definition for LOD.
A consistent error evaluation is required for all provided quantities.
The explanation of the enhanced charge transfer is rudimentary and not satisfying. On page 6 (page numbers are mixed in the manuscript!) line 193 it is written that “improvement of electronic contact between the electroactive centers and electrode surface. The probable reason for this phenomenon are the negative charges of the ssDNA strand.” This should be explained in more detail for instance in terms of energy states, charge transfer probability, and reorganization energy. It furthermore remains unclear why the current changes when the DNA hybridizes (this is the actual principle of the detection scheme). Does the number of surface-bound redox complexes that are involved in the charge transfer varies? Is this related to alternations of the complex stability during DNA hybridization?
Annotations of minor importance:
A considerable number of typos in the manuscript require careful proofreading.
Please specify which buffer was used for the electrochemical investigations.
Author Response
Please, see the attachment

Round 2
Reviewer 2 Report
I recommend the manuscript to be published.
Reviewer 3 Report
The manuscript can be published after the LOD is determined according to IUPAC definition and the errors for the values in table 4 and 5 had been estimated